# Efficacy of Praziquantel in Treating *Schistosoma haematobium* Infection Among Ethiopian Children

**DOI:** 10.3390/biomedicines12112463

**Published:** 2024-10-27

**Authors:** Louis Fok, David M. Brett-Major, Berhanu Erko, John Linville, Hongying Daisy Dai, Yohannes Negash, Abebe Animut, Abraham Degarege

**Affiliations:** 1Department of Epidemiology, University of Nebraska Medical Center, Omaha, NE 68198, USA; louis.fok@unmc.edu (L.F.); david.brettmajor@unmc.edu (D.M.B.-M.); 2Division of Infectious Diseases, College of Medicine, University of Nebraska Medical Center, Omaha, NE 68198, USA; 3Aklilu Lemma Institute of Pathobiology, Addis Ababa University, Addis Ababa P.O. Box 1176, Ethiopia; berhanu.erko@aau.edu.et (B.E.); yohannes.negash@aau.edu.et (Y.N.); abebe.animut@aau.edu.et (A.A.); 4Department of Environmental, Agricultural & Occupational Health, University of Nebraska Medical Center, Omaha, NE 68198, USA; john.linville@volunteer.unmc.edu; 5Department of Biostatistics, University of Nebraska Medical Center, Omaha, NE 68198, USA; daisy.dai@unmc.edu

**Keywords:** *Schistosoma haematobium*, urogenital schistosomiasis, praziquantel, treatment, efficacy, egg reduction

## Abstract

**Background/Objectives**: Praziquantel is a cornerstone of schistosomiasis control and elimination efforts. Continued surveillance of praziquantel efficacy is needed to monitor for the development of resistance, as well as to help public health officials gauge the effect of mass praziquantel administration on schistosomiasis control in communities, since it is the only drug used in schistosomiasis control programs. The objective of this study was to assess the praziquantel cure rate and egg reduction rate against urogenital schistosomiasis. **Methods:** This study enrolled 977 children from 12 villages in Afar and Gambella, Ethiopia, who provided urine samples that were checked for *Schistosoma haematobium* infection at baseline using urine filtration microscopy. Infected individuals were provided a single dose of praziquantel (40 mg/kg body weight) and retested six weeks post-treatment. **Results:**
*S. haematobium* was recovered from baseline urine specimens in 177 of 977 (18%) participating children. One hundred six of these children completed therapy and presented for subsequent evaluation at six weeks; 91 children were egg-free. The egg reduction rate was 97%; changes in egg burden among the 15 children who did not achieve cure varied widely. Cure rates were better among children with light-intensity infections. No significant differences in egg reduction rates were found based on the demographic variables examined. **Conclusions:** Standard praziquantel monotherapy remains an effective treatment against urogenital schistosomiasis in Ethiopia.

## 1. Introduction

Schistosomiasis is considered to be a devastating parasitic infection afflicting humans after malaria [1,2]. Although it claims the lives of thousands worldwide every year, its human cost is measured more in terms of disability than death. Growth retardation, cognitive decline, and bladder cancer are among some of the many complications caused by infection with *Schistosoma* occurring among the hundreds of millions with human schistosomiasis worldwide [3,4]. Continued efforts to improve control programs and treat infected individuals are critical.

For more than four decades, praziquantel chemotherapy has formed the backbone of schistosomiasis control and treatment [5]. Although the drug’s mechanism of action is unknown, it is believed that praziquantel’s primary anti-schistosomal properties derive from its ability to interfere with calcium channels in trematodes [6], in addition to disrupting other essential molecules in worms [7]. The drug is easy to administer orally and is generally well-tolerated, with mild adverse effects [8,9], most of which are headache, abdominal pain, and vertigo [10]. Although praziquantel has served its purpose well, there is a concern that resistance to the drug develops in some settings [7,8]. It remains the only drug approved for the treatment of schistosomiasis. Cure rates differ across species and regions [11,12,13]. All of these provide an impetus for monitoring drug efficacy in support of improved patient outcomes and appropriate public health programming.

Of the six *Schistosoma* species known to infect humans, *S. haematobium* is the only one that primarily causes urogenital schistosomiasis [3,14]. The distribution of the species spreads across Africa and the Middle East and, to a lesser extent, the islands of the Indian Ocean, with contemporary examples of brief reintroduction in sub-tropical zones in Europe [15]. Ethiopia is among the countries heavily affected by the disease [16].

While some studies have been conducted in various parts of Ethiopia to examine the efficacy of standard praziquantel therapy in treating urogenital schistosomiasis [9,17], there has been no published data based on the population in Afar, Ethiopia, since 1988 [17] and none on the population of Gambella to date. The prevalence of *S. haematobium* infection among children in some villages of the Afar and Gambella regions reaches 37.8% and 43.8%, respectively [18,19]. These two regions of Ethiopia are important for studying schistosomiasis treatment due to their distinct characteristics and the challenges in accessing these remote populations. The Afar people are nomadic pastoralists, whose way of life centers around raising livestock [20], and they often work in areas flooded with waters that transmit schistosomes [21]. For decades, they have also faced recurring famines [22]. In contrast, the people of Gambella, while traditionally more sedentary, have become increasingly transient due to the influx of refugees fleeing the conflict in South Sudan [23]. This region also has one of the highest prevalence rates of *S. haematobium* infection in the entire country [16]. Moreover, the people of Afar and Gambella share a long history of conflict, war, and dispossession [24,25]. As a result, little is known about the efficacy of praziquantel in curing *S. haematobium* infection in these populations.

Additionally, there are limited data on the efficacy of Bermoxel, a praziquantel brand manufactured by Medochemie in Cyprus, in treating *S. haematobium* infections. Most existing evidence is based on praziquantel previously marketed under the brand name Biltricide by the Bayer company in Germany [9] and later as Cesol by Merck in the same country [26,27,28,29]. Furthermore, a previous study raised concerns about Bermoxel, reporting that it did not meet the dissolution standards set by the United States Pharmacopeia [30]. This highlights the need to generate real-world evidence of Bermoxel’s in vivo performance. To address these knowledge gaps and ensure ongoing monitoring of the drug’s real-world effectiveness in endemic populations, we have evaluated the cure and egg reduction rates of a single-dose praziquantel treatment (40 mg/kg, Bermoxel brand) for urogenital schistosomiasis in school-aged children in the Afar and Gambella regions of Ethiopia.

## 2. Materials and Methods

This study was conducted as part of a trial examining the use of pooled urine sample testing as a potentially cost-effective method for diagnosing infections at the community level [31].

### 2.1. Study Design, Population, and Area

In July 2023 and July 2024, children between the ages of five and fifteen residing in rural villages in the Awash Valley of Afar, Ethiopia, and Gambella, Ethiopia, were invited to participate in this unblinded prospective study. Villages were selected based on prior data [16,18,32] to maximize the number of positive cases, as well as in collaboration with administrative officials and local guides, who took accessibility and local control programming activities into consideration. The invited children were screened for inclusion. Children were eligible if they had parental consent and resided in a village that did not receive mass praziquantel administration within the last 90 days.

### 2.2. Sample Size Determination

The sample size of this supplementary study was determined by the parent study, whereby all of the study participants were included.

### 2.3. Data Collection

Enrollment and data collection took place in 12 villages during the late mornings and early afternoons. Once enrolled, biographical details were recorded, including age, sex, village in which they reside, and a sequentially assigned number to identify each urine sample to an individual child. The children who were included were then instructed to provide at least 100 mL of urine in a plastic jar with a marked level line. Children who were initially unable to provide a sufficient urine volume were provided with water to drink. Formaldehyde was added to the collected samples. The sample jars were then sealed with a lid and labeled with the child’s identification number.

### 2.4. Processing of Urine Samples

Urine samples were immediately tested in the field with urine reagent strips to detect hematuria. A sample aliquot of 10 mL was rapidly filtered onto a polycarbonate membrane and examined under a microscope. If either of these two methods rendered a positive result, the child and parents were provided with instructions and a single oral praziquantel (*Bermoxel* by Medochemie, Limassol, Cyprus) dose of 40 milligrams per kilogram of body weight. Thereafter, the remaining urine was transported to the Medical Parasitology Laboratory of Aklilu Lemma Institute of Pathobiology, Addis Ababa University, for further processing. For each sample, 10 mL was examined for the presence and count of *S. haematobium* eggs using standard urine filtration microscopy (UFM). Any remaining children whose samples turned out positive but had not been treated during the baseline data collection were sought after for treatment approximately six weeks after the baseline when enrolled children who were treated at baseline were invited for re-testing using the same laboratory UFM methodology.

### 2.5. Data Analysis

This study performed an intent-to-treat analysis. Only the results processed in the laboratory were used to categorize the children’s infection status and egg counts. Cure and egg reduction rates were assessed overall and stratified based on demographic variables and infection intensity as defined by baseline egg counts. If the baseline egg count was between 1 and 49, the child was categorized as having a light-intensity infection, whereas having an egg count greater than 49 would have the child categorized as having a heavy-intensity infection [33]. The cure rate was calculated using the formula 100 × (# of children testing negative at follow-up) ÷ (# of children testing positive at baseline). The egg reduction rate was calculated using the formula 100 × [1 − (average egg count among all persons at follow-up) ÷ (average egg count among all positive persons at baseline)]. All other estimations were generated using SAS version 9.4 (SAS Institute, Cary, NC, USA).

The children’s infection intensities were stratified based on age group (5–10 and 11–15), sex (female vs. male), and village. Because many of the egg counts were less than five and highly uneven among sub-groups, Fisher’s exact test was conducted, and corresponding *p*-values were estimated using an alpha level of 0.05 to determine if there were statistically significant differences in the distribution of infection intensities based on age, sex, and village. 

Results stratified based on infection intensity, age, sex, and village were tested to detect differences in average egg counts (at baseline and at follow-up), cure rates, and egg reduction rates. Specifically, an independent samples t-test was performed to examine potential differences in the average egg count at baseline by sex and age groups; ANOVA was used to compare the mean egg count of children at baseline based on the villages where they live. McNemar’s test was performed to compare the cure rates by infection intensity at baseline, sex, and age group; a Cochran’s Q Test was performed to test the difference in cure rates by villages. To determine differences in average egg count at follow-up when stratified by infection intensity, sex, and age group, an independent samples t-test was performed; when stratified by village, an ANOVA was performed. A paired t-test was performed to determine differences in egg reduction rate when stratified by infection intensity, sex, and age; a repeated measures ANOVA was performed to compare egg reduction rates by villages.

### 2.6. Ethical Approval

The research team obtained approval to conduct this study from the Institutional Review Boards of the University of Nebraska Medical Center (IRB # 908-19-EP) and Aklilu Lemma Institute of Pathobiology, Addis Ababa University (Ref. No. ALIPB-IRB/10/2012/20).

## 3. Results

Totals of 572 and 405 children from Afar and Gambella were examined, respectively. Of the 977 children examined overall, 216 were positive at baseline. Of these, 177 received oral praziquantel at a single dose of 40 mg per kilogram of body weight. One hundred six of the children returned for re-testing post-treatment.

### 3.1. Baseline Infection Status

Of the 106 children who presented for post-treatment evaluation, 88 (83%) were classified as having a light-intensity infection (see Table 1). Six girls and 12 boys had heavy infections, 16 of which were among younger children (those aged 5 through 10). A total of 55% were males (N = 58). In stratified univariate analysis, no statistically significant difference was found in the distribution of infection intensities by sex (*p* = 0.31). However, infection intensity differed significantly by village (*p* < 0.01) and age (*p* = 0.03), with younger children having a greater prevalence of heavy infections.

Boys had greater average egg counts at baseline than girls (38 vs. 26 eggs per 10 mL—“EPU”), but the finding did not reach statistical significance (*p* = 0.30) (see Table 2). There were statistically significant differences in average EPU counts between age groups (*p* = 0.04), markedly so between villages, ranging from 1 to 98 (*p* < 0.01).

### 3.2. Cure Rates and Egg Reduction Rates

Ninety-one of the 106 children (86%) who were treated for *S. haematobium* with praziquantel and available for assessment at six weeks post-treatment were cured (see Table 2). The cure rate was lower in those with heavy infections than those with light infections (67% vs. 90%, *p* < 0.01). There was, however, no statistically significant difference in cure rates between sexes (*p* = 0.16), age groups (*p* = 1.00), and villages (*p* = 0.92). The total egg reduction rate was 97%. Among the 15 children who did not achieve a cure, six-week egg count differences from baseline varied widely, from >99% to an increase in counts by a third. There were no statistically significant differences in egg reduction rates among infection intensities, sexes, ages, and villages.

## 4. Discussion

In our cohort, single-dose praziquantel therapy at a 40 mg/kg body weight demonstrated an overall cure rate of 86% and an egg reduction rate of 97%. The cure rate results exceed those from a meta-analysis of 47 trials that incorporated data from across Africa and Egypt, yielding a pooled cure rate of 74% [8], analogous to what we observed in high-intensity infections in Afar, Ethiopia. Another meta-analysis focusing on 15 schistosomiasis studies in Ethiopia had pooled cure and egg reduction rates of 94% and 85%, respectively [34]. Most importantly, the egg reduction rate of 97% in this study is within the cutoff values (≥90%) recommended by the WHO [35], demonstrating that the drug remains effective and without signs of resistance in the study area.

Differences in the influence of various demographic factors on cure and egg reduction rates did not reach statistical significance. However, there was a strong trend toward a better cure rate among female school-aged children than that of their male counterparts (92% vs. 81%), which could be impacted by a mild trend toward more light infections in female children. This finding mirrors that of a study in Gabon, which found that praziquantel efficacy differed across sexes, with the lowest among males (68%) and the highest among females (90%) [36]. That study, however, had a longer period prior to re-assessment (3 months) and so may be more vulnerable to behavioral differences that lead to reinfection. After all, some research estimates that 14% of Ethiopian children are reinfected with schistosomiasis within six months post-treatment [37]. Ideally, this study should have employed a longer follow-up time with multiple post-treatment tests to distinguish between reinfection and treatment failure between girls and boys. Ultimately, this study’s inability to do so due to resource constraints proves to be a limitation on our capacity to discern why males had a lower cure rate than females. Nonetheless, the possibility of an impact from sex-dependent pharmacodynamics or other biological factors cannot be excluded from either study. The fact that infection intensity was greater in certain villages, as well as in younger children, is consistent with the role of shared exposures and the number of opportunities to interact with school-based treatment programs.

The main limitation of this study is the potential for bias due to loss to follow-up. Of the 177 children who tested positive at baseline and were administered praziquantel, 71 were unable to be located six weeks after treatment. This may be due in part to the culture and nomadic nature of the Afar people. Indeed, studies from the Ethiopian context have uncovered that children living in rural areas [38], youngsters [39,40], those with poor medication adherence [38,39], and those living far away from healthcare services [40] are at high risk of being lost to follow-up. As this study almost exclusively studied persons living in rural areas and those living far from healthcare facilities, on top of the fact that the vast majority of included children were ten years of age or younger, it is understandable why there was a large loss to follow-up. If it were the case that those who were lost to follow-up were less likely to be cured, this study would be biased away from the null. Furthermore, despite our study’s sample size exceeding the recommended minimum number—50—for conducting anthelminthic efficacy studies [35], the sample size was too small to discern any differences in results among different groups of persons. Future studies that seek to compare the prevalence of infection intensities between villages should recruit larger sample sizes. This study did not examine medication adherence, so a per-protocol analysis was not performed. Low medication adherence would bias results toward the null.

It is worth noting that the parent study analyzed the performance of a differential filtration device, FlukeCatcher (FC), at improving the detection of schistosome eggs in the children’s urine. The results of the study confirmed the superiority of FC over the standard UFM in detecting *S. haematobium* eggs [31]. As a result, UFM produced a significant number of false negatives, particularly among individuals with very low-intensity infections. These cases were excluded from the study analysis because FC was not used to retest the samples after praziquantel treatment. It is not known exactly how the results of the present study would have been affected had the more sensitive FC been used as a diagnostic tool both at baseline and post-treatment. While the inclusion of more light-intensity infections that could be detected through FC at baseline may have increased the estimated cure rate, it is possible that FC could also detect light-intensity infections after treatment, which could have been missed by UFM, thereby decreasing the estimated cure rate. Future studies should employ additional diagnostic tools, such as FC, to more comprehensively detect cases of *S. haematobium* infection for drug efficacy evaluation.

Moreover, this study only evaluated a single-dose regimen, as opposed to a double-dose regimen, which has been linked to higher ERRs [41]. This makes gradual changes in *in vivo* drug resistance to praziquantel difficult to ascertain from tracking ERR alone. There is a lack of contemporary information on the impact of current approaches on patient outcomes at the individual level and reinfection and onward transmission at the community level. Opportunities for innovation and more proactive risk assessment remain in schistosomiasis risk management.

To the best of our knowledge, this is the first known study to evaluate the efficacy of praziquantel within the Gambella region and the first such study in three decades to do the same in Afar. Despite their unique circumstances and cultures, the underserved and displaced children of Afar and Gambella, Ethiopia, can be assured that praziquantel is effective against urogenital schistosomiasis infections. Furthermore, there is limited information on the efficacy of Bermoxel against schistosomiasis infection *in vivo*. The current findings address this gap by confirming the effectiveness of the Bermoxel praziquantel brand in treating *S. haematobium* infections. In light of this finding, *Bermoxel* has proven to be worthy of consideration for mass drug administration (MDA) programs. Were it to be used as the brand of choice for such programs, it would be expected to cure the overwhelming majority of infected children and significantly reduce the egg burden among those who are not cured. As mentioned earlier, a double dose of *Bermoxel* could prove to reduce egg counts even further and ensure *S. haematobium* control in endemic regions.

Given praziquantel’s effectiveness, these results suggest that resistance does not appear to be of significant public health concern in these regions of Ethiopia. Prior research has found schistosome isolates in Egypt (in particular the regions surrounding the Nile River) [42,43] and Senegal [43] that demonstrate increased tolerance to praziquantel. In addition, praziquantel resistance has been artificially induced through gain-of-function research under laboratory conditions [42]. Although praziquantel resistance has been identified in some schistosome isolates from endemic areas and under laboratory conditions, it is not currently considered clinically significant [44]. Furthermore, there is no evidence that large-scale praziquantel resistance has developed in Africa as a result of MDA programs [45]. This study contributes to this body of knowledge by affirming that praziquantel resistance is not present in the Afar and Gambella regions of Ethiopia. Indeed, research suggests that most cases of persistent schistosomiasis are due to reinfection [46] or heavy infections resulting from intense transmission within endemic areas [47]. However, the possibility of praziquantel resistance developing in the future is real [44] and warrants continued monitoring of the drug’s efficacy [11], especially in light of the reality that there are no alternative anthelmintics for fighting schistosomiasis [5].

## 5. Conclusions

Single-dose praziquantel monotherapy at a dose of 40 mg/kg is an effective treatment against urogenital schistosomiasis among school-aged Ethiopian children that dramatically reduces the egg burden among those who do not achieve clinical cure. Its efficacy among Ethiopian children is non-inferior to that demonstrated in other populations. Praziquantel’s efficacy against schistosomiasis should be continually assessed to monitor for any possible emerging resistance to the drug.

## Figures and Tables

**Table 1 biomedicines-12-02463-t001:** Baseline infection intensities by demographic variables.

		Infection Intensity		*p*
		Light	Heavy	
Age	5–10	55 (77%)	16 (23%)	0.03
	11–15	33 (94%)	2 (6%)	
Sex	Female	42 (88%)	6 (12%)	0.31
	Male	46 (79%)	12 (21%)	
Village	Andada	6 (100%)	0 (0%)	<0.01
	Abajo	9 (50%)	9 (50%)	
	Buri	13 (100%)	0 (0%)	
	Erinbirta	1 (100%)	0 (0%)	
	Gabole	7 (88%)	1 (12%)	
	Haledebe	6 (100%)	0 (0%)	
	Iore	0 (0%)	1 (100%)	
	Kalat	3 (100%)	0 (0%)	
	Kusra	2 (50%)	2 (50%)	
	Mender 17	26 (100%)	0 (0%)	
	Rebada	1 (100%)	0 (0%)	
	Tegni	14 (74%)	5 (26%)	
Total		88 (83%)	18 (17%)	

**Table 2 biomedicines-12-02463-t002:** Efficacy of praziquantel against *Schistosoma haematobium* stratified by sex, age, village, and infection intensity.

		At Baseline					At Follow-Up	
		Number of Positives	Average Egg Count		Number Cured	Cure Rate	Average Egg Count	Egg Reduction Rate
Infection intensity	Light	88	13		79	90%	1	94%
	Heavy	18	130		12	67%	2	99%
	*p*-value		< 0.01		<0.01		0.21	0.21
Sex	Female	48	26		44	92%	1	98%
	Male	58	38		47	81%	1	96%
	*p*-value		0.30		0.16		0.10	0.19
Age	5–10	71	39		61	86%	1	98%
	11–15	35	20		30	86%	1	95%
	*p*-value		0.04		1.00		0.79	0.38
Village	Andada	9	3		6	67%	0	100%
	Abaro	18	98		16	89%	1	99%
	Buri	17	14		11	65%	1	92%
	Erinbirta	1	3		1	100%	0	100%
	Gabole	9	24		6	67%	2	92%
	Haledebe	14	8		6	43%	0	100%
	Iore	1	57		1	100%	0	100%
	Kalat	3	19		3	100%	0	100%
	Kusra	11	51		3	27%	1	99%
	Mender 17	26	12		22	85%	1	89%
	Rebada	2	4		1	50%	0	100%
	Tegni	19	36		15	79%	1	97%
	*p*-value		<0.01		0.33		1.00	0.99
Overall		106	32.7		91	86%	1.0	97%

## Data Availability

The data presented in this study are available on request from the corresponding author. The data are not publicly available due to privacy and ethical concerns.

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
