# Peer review of "Efficacy of Praziquantel in Treating Schistosoma haematobium Infection Among Ethiopian Children"

_biomedicines, 2024, doi:10.3390/biomedicines12112463_

Round 1
Reviewer 1 Report (Previous Reviewer 1)
Comments and Suggestions for Authors
Authors edited the manuscripts accordingly.
Reviewer 2 Report (Previous Reviewer 2)
Comments and Suggestions for Authors
I believe the authors have successfully addressed the key points in their work. They have highlighted the gaps in access to anti-schistosoma drugs within African society, which adds significant value to the study. By drawing attention to these critical issues, the paper has been enriched, making it a valuable contribution to the field. Given these improvements, the manuscript is well-suited for publication in its current form.
This manuscript is a resubmission of an earlier submission. The following is a list of the peer review reports and author responses from that submission.
Round 1
Reviewer 1 Report
Comments and Suggestions for Authors
The current study aims to asses the efficacy of Paraziquantel for urogenital schistosomiasis among Ethiopian children. The manuscript is clear and well-structured. the cited references are relevant and updated. the manuscript's results are reproducible based on the details given in methods section. the tables are appropriate and understandable. the conclusions are consistent with evidence and arguments presented. the ethics state is adequate. I have only one comment on introduction. I think authors need to state the mechanism and side effects of praziquantel.
Reviewer 2 Report
Comments and Suggestions for Authors
I reviewed the manuscript entitled “Efficacy of praziquantel for the treatment of S. haematobium infection among Ethiopian children”. In the present study, it is stated that Praziquantel remains effective in treating urogenital schistosomiasis, as demonstrated by a study involving 977 children from Ethiopia, with an egg reduction rate of 97%. Of the 177 infected children, 91 were cured after a single dose, with better cure rates observed in those with lighter infections, indicating that praziquantel is still a reliable treatment in schistosomiasis control efforts.
Although the authors conducted the study and achieved satisfactory results, the study does not contribute new or particularly interesting insights to the existing body of knowledge. According to well-established sources, such as the CDC, Praziquantel is already the recommended treatment for schistosomiasis, with clearly defined dosage and treatment duration guidelines. Therefore, while the study confirms the drug's efficacy, it does not provide novel findings that significantly advance current understanding or treatment practices.
In addition, the number of cited references could be expanded to strengthen the study's foundation. The article relies heavily on references from authoritative organizations such as the WHO and CDC, which, while credible, limits the diversity of sources. Including a broader range of studies and research papers would enhance the depth and credibility of the findings, offering a more comprehensive view of the topic.
Reviewer 3 Report
Comments and Suggestions for Authors
The article by Fok et al. provides a thorough assessment of the efficacy of praziquantel in treating urogenital schistosomiasis in Ethiopia, contributing to ongoing efforts to monitor and treat this parasitic disease. The study focused on children in two regions of Ethiopia (Afar and Gambella). It showed promising results regarding cure rates and egg count reduction after a single dose of praziquantel, confirming the drug's efficacy in these populations.
The study fills a significant gap in the literature, as no recent data have been collected from these specific regions of Ethiopia (Afar since 1988 and none from Gambella). Schistosomiasis remains a significant public health problem in sub-Saharan Africa, and this study helps to update the data on the efficacy of praziquantel, which is crucial for disease management and potential monitoring of drug resistance.
The study design is well done, and results are presented clearly in two tables. The study includes 977 children from 13 villages and uses urine filtration microscopy, a reliable method for detecting Schistosoma haematobium eggs. The statistical analysis of the study, including t-tests and ANOVA for different subgroups, is rigorous, with sufficient attention paid to demographic variables such as age, sex, and infection intensity.
The cure rate (86%) and egg reduction rate (97%) indicate that praziquantel remains highly effective in treating urogenital schistosomiasis in the study area. These rates are consistent with WHO recommendations for drug efficacy and comparable with results from other endemic regions, confirming the continued utility of the drug in schistosomiasis control programs.
The authors balance the discussion of potential limitations of the study, such as the loss of follow-up (71 children) and the inability to account for reinfection or adherence to praziquantel dosing. Acknowledging these limitations enhances the credibility of the study by pointing to areas for further research.
There are, however, some areas for improvement:
1. While the study's sample size is generally adequate, the significant proportion of children lost to follow-up (40%) may introduce bias. The authors acknowledge this, but a deeper investigation into the reasons for this loss (e.g., the nomadic lifestyle of some participants) could improve the study's conclusions.
2. The study does not investigate this further, although reinfection is mentioned as a potential problem. Given the high rates of reinfection commonly associated with schistosomiasis in endemic areas, it would be beneficial to consider longer follow-ups to distinguish between treatment failure and reinfection.
3. Although the study addresses the issue of potential praziquantel resistance, it could have benefited from a more detailed discussion of early indicators of resistance or emerging research in this area. Although the results are promising, the broader context of praziquantel resistance in Africa would have added depth to the article.
4. The study relies primarily on traditional methods, such as urine filtration microscopy and single-dose treatment, which are standard but not innovative. Exploring newer diagnostic technologies or treatment protocols may provide more comprehensive insight into the future of schistosomiasis control.
Overall, the article makes a valuable contribution to understanding the efficacy of praziquantel in Ethiopia, confirming its continued utility in controlling urogenital schistosomiasis in school-age children. The results of the study fit well with previous studies, although its limitations, particularly in terms of follow-up and sample size, suggest areas for improvement.